# Al_2_O_3_-Based Hollow Fiber Membranes Functionalized by Nitrogen-Doped Titanium Dioxide for Photocatalytic Degradation of Ammonia Gas

**DOI:** 10.3390/membranes12070693

**Published:** 2022-07-06

**Authors:** Edoardo Magnone, Jae Yeon Hwang, Min Chang Shin, Xuelong Zhuang, Jeong In Lee, Jung Hoon Park

**Affiliations:** Department of Chemical and Biochemical Engineering, Dongguk University, 30, Pildong-ro 1 gil, Jung-gu, Seoul 04620, Korea; magnone.edoardo@gmail.com (E.M.); fzytm@naver.com (J.Y.H.); gogokill31@naver.com (M.C.S.); zhuangxuelong@dgu.ac.kr (X.Z.); jungin0407@naver.com (J.I.L.)

**Keywords:** inorganic membrane, photocatalytic membrane reactor, separation process, membrane applications, air purification, titanium dioxide, gaseous ammonia (NH_3_) degradation

## Abstract

In recent years, reactive ammonia (NH_3_) has emerged as a major source of indoor air pollution. In this study, Al_2_O_3_-based hollow fiber membranes functionalized with nitrogen-doped titanium dioxide were produced and successfully applied for efficient heterogeneous photocatalytic NH_3_ gas degradation. Al_2_O_3_ hollow fiber membranes were prepared using the phase inversion process. A dip-coating technique was used to deposit titanium dioxide (TiO_2_) and nitrogen-doped titanium dioxide (N-TiO_2_) thin films on well-cleaned Al_2_O_3_-based hollow fiber membranes. All heterogeneous photocatalytic degradation tests of NH_3_ gas were performed with both UV and visible light irradiation at room temperature. The nitrogen doping effects on the NH_3_ heterogeneous photocatalytic degradation capacity of TiO_2_ were investigated, and the effect of the number of membranes (30, 36, 42, and 48 membranes) of the prototype lab-scale photocatalytic membrane reactor, with a modular design, on the performances in different light conditions was also elucidated. Moreover, under ultraviolet and visible light, the initial concentration of gaseous NH_3_ was reduced to zero after only fifteen minutes in a prototype lab-scale stage with a photocatalytic membrane reactor based on an N-TiO_2_ photocatalyst. The number of Al_2_O_3_-based hollow fiber membranes functionalized with N-TiO_2_ photocatalysts increases the capacity for NH_3_ heterogeneous photocatalytic degradation.

## 1. Introduction

Ammonia is an inorganic gas pollutant with the molecular formula NH_3_ that is commonly found in low concentrations in the indoor environment. The environmental effects of odor emissions from livestock structures were also identified as critical issues [1]. For such low concentrations of NH_3_, adsorption and ventilation techniques are not cost-effective [2]. Other alternative methods, such as oxidation processes combining ultraviolet irradiation with ozone [1], and a reactor consisting of a set of two-stage-in-series biotrickling filters, an influent gas supply system, and a liquid recirculation system [3], have recently been proposed. A new inexpensive approach is required to eliminate this pollutant and maintain a clean indoor environment. In order to make progress in this field, it is necessary to investigate a new photocatalytic membrane reactor and demonstrate that there is a substantial environmental benefit to be gained from reducing the gaseous NH_3_ pollutant.

Titanium dioxide (TiO_2_) and nitrogen-doped titanium dioxide (N-TiO_2_) materials have gained prominence as the most researched semiconductor materials for photocatalytic purposes, including their use in devices for photocatalytic degradation of gaseous ammonia under various light sources [4,5,6,7,8,9,10,11].

Geng et al. in 2008 investigated the photocatalytic degradation of gaseous NH_3_ by using nano-TiO_2_ photocatalyst supported on latex paint film under UV irradiation [5]. From the kinetic point of view, it was found that the photocatalytic degradation of ammonia follows a pseudo-first order reaction [5,6,7,8].

Wang et al. reported that the removal percentage of NH_3_ after about a 9 h photocatalytic reaction under visible light irradiation reached 53.1% using the optimal TiO_2_ thin film doped with iron (III) [7]. After that, in 2013 Zendehzaban et al. coated TiO_2_ on light expanded clay aggregate granules (LECA)—which is a porous and lightweight support—and found that more than 85% of NH_3_ was photocatalytically removed within 300 min of the process under UV irradiation [8].

Li et al. reported in 2018 that a hierarchical-structured composite, ultrafine TiO_2_ encapsulated in a nitrogen-doped porous carbon framework, can be used as a photocatalyst to degrade ammonia gas [10]. This is an interesting case in which the photocatalytic activity was excellent with 100% efficiency; it is represented by ultrafine TiO_2_ encapsulated in a nitrogen-doped porous carbon framework [10]. The authors attributed the superior photocatalytic performance of this photocatalytic material to its large surface area and abundant pore structure [10].

Recently, Čižmar et al. studied the photocatalytic activity of nanostructured Cu-modified vertically aligned TiO_2_ nanotube arrays in a mini-photocatalytic wind tunnel reactor (MWPT) [11].

However, despite various studies on heterogeneous photocatalytic NH_3_ gas degradation through the use of different photocatalytic materials [1,2,3,4,5,6,7,8,9,10,11,12,13] and relatively few tentative attempts to investigate alternative ways to obtain a compact photocatalytic membrane reactor for indoor pollution abatement [11], to the best of our knowledge, there has been no case applied to the heterogeneous photocatalytic degradation process using Al_2_O_3_-based hollow fiber membranes functionalized by N-TiO_2_ in a novel photocatalytic membrane reactor to completely reduce the NH_3_ pollutant concentration.

In this study, Al_2_O_3_-based hollow fiber membranes functionalized by N-TiO_2_ were investigated as an advanced photocatalyst for heterogeneous photocatalytic degradation of an NH_3_ gas pollutant in a novel prototype lab-scale photocatalytic membrane reactor, with a modular design, based on 30, 36, 42, and 48 membranes under different light sources such as UV and visible light irradiation.

This paper reveals for the first time, to the authors’ knowledge, the potential of the developed prototype lab-scale photocatalytic membrane reactor with Al_2_O_3_-based hollow fiber membranes functionalized by N-TiO_2_ as a promising and attractive candidate to remove NH_3_ pollutants and then maintain a clean indoor environment. In addition, comparative studies of the NH_3_ photocatalytic degradation processes were carried out using the undoped TiO_2_ photocatalysts.

## 2. Materials and Methods

### 2.1. Materials

The materials and solvents used in the phase inversion process to prepare the Al_2_O_3_ hollow fiber membrane and the dip-coating deposition of TiO_2_ films were used exactly as received without further purification and are detailed in the Appendix A.

### 2.2. Preparation of Al_2_O_3_ Hollow Fiber Membranes

Al_2_O_3_ hollow fiber membranes were produced in two steps: (a) phase inversion spinning and (b) high-temperature sintering. In brief, a polymer solution including 33.5 wt% of NMP as a solvent and 6 wt% of PVP as a binder were mixed together. Al_2_O_3_ powder was added into the polymer solution with 0.5 wt% of PVP as a dispersing agent. To determine the composition of the suspension, we used previous literature results [14,15]. In detail, the prepared suspension of Al_2_O_3_ powder (60%) and additives (40%) was stirred (150 rpm) for one day before being degassed with a vacuum pump (IDP3, Varian, Palo Alto, CA, USA) for about one hour. The mixed and degassed suspension was extruded through an iron nozzle at a pressure of three bars. The air gap was set at 10 cm. Deionized water was used as a coagulant at room temperature. The phase transition process was completed within one day. Finally, Al_2_O_3_ hollow fiber membranes were dried at 100 °C for 12 h before being sintered at 1300 °C for 3 h.

### 2.3. Preparation of Undoped TiO_2_ and N-TiO_2_ Photocatalyst Powders

The sol-gel method was used to produce the photocatalysts under consideration. As a titanium precursor, titanium (IV) isopropoxide (TTIP) was used. A total of 48 g of TTIP were mixed with 150 mL of deionized water and stirred for 1 h at room temperature using magnetic stirrers. A filtered white slurry was dried at 110 °C for 12 h to remove the residual water. Finally, the obtained photocatalyst powder was calcined for 3 h at 400 °C. Urea was used as a nitrogen source in the synthesis of N-TiO_2_ [12]. In this case, 50 g of urea were previously dissolved in deionized water and stirred at room temperature for 1 h. Dropwise additions of TTIP were made, and then the solution was stirred for another hour. The calcination procedure used to prepare N-TiO_2_ was the same as that used to prepare undoped TiO_2_. The powders appear to be different colors at first glance (see Appendix A).

### 2.4. Preparation of Al_2_O_3_-Based Hollow Fiber Membranes Functionalized by Undoped TiO_2_ and N-TiO_2_ Photocatalysts

Undoped TiO_2_ and N-TiO_2_ photocatalysts were applied on Al_2_O_3_-based hollow fiber membranes by a dip-coating process based on tetraethyl orthosilicate (TEOS) solution as a silica-based binder [16]. In brief, 21 g of the photocatalyst under consideration were dispersed in 78 g of ethanol with 8 g of silica-based binder solution. The suspension was stirred for 1 h at room temperature to obtain a homogenous photocatalyst coating solution. The Al_2_O_3_-based hollow fiber membranes were cleaned in acetone under supersonic conditions and dried at 100 °C before immersing them in the photocatalyst coating solution for 1 h. Then, after this immersion time, the Al_2_O_3_-based hollow fiber membranes functionalized by undoped TiO_2_ and N-TiO_2_ photocatalysts were washed with common deionized water and dried at 150 °C for 12 h.

### 2.5. Characterization of Undoped TiO_2_ and N-TiO_2_ Photocatalyst Powders and Deposited Films

X-ray diffraction (XRD, Ultima IV, Rigaku, Tokyo, Japan), an accelerated surface area and porosimetry analyzer (ASAP 2010 Instrument, Micromeritics Instrument Corporation, Atlanta, GA, USA), a transmission electron microscope (TEM, JEM-F200, JEOL Ltd., Tokyo, Japan), X-ray photoelectron spectroscopy (XPS, Veresprobe II, ULVAC-PHI, Chigasaki, Japan), scanning electron microscopy (SEM, model S-4800, Hitachi, Tokyo, Japan) with an energy dispersive spectrometer (EDS, model S-4800, Hitachi, Tokyo, Japan), a UV-Vis diffuse reflectance spectrophotometer (UV-Vis DRS, SolidSpec-3700, Shimadzu, Kyoto, Japan), and photoluminescence spectroscopy (PL, LabRAM HR-800, Horiba. Ltd., Kyoto, Japan) were employed to characterize not only the undoped TiO_2_ and N-TiO_2_ photocatalyst powders but also the Al_2_O_3_-based hollow fiber membranes functionalized by undoped TiO_2_ and N-TiO_2_ photocatalysts.

### 2.6. Photocatalytic Degradation of Gaseous Ammonia

A novel prototype lab-scale photocatalytic membrane reactor, with a modular design, was specifically designed to evaluate the NH_3_ heterogeneous photocatalytic degradation capacity of the Al_2_O_3_-based hollow fiber membranes functionalized by undoped TiO_2_ and N-TiO_2_ photocatalysts in the application field of air-purifying filters and indoor pollution abatement. The reactor was designed to also include a brushless DC fan to push the gas toward the cylindrical photocatalytic membrane chamber. The acrylic chamber was designed to contain a different number of functionalized membranes in order to study the geometrical effects on the NH_3_ heterogeneous photocatalytic degradation performances under different light sources such as UV and visible light irradiation. The modular photocatalytic membrane reactors were made of 30, 36, 42, and 48 Al_2_O_3_-based hollow fiber membranes functionalized by undoped TiO_2_ and N-TiO_2_ photocatalysts, with membrane surface areas of 8.430 × 10^−3^, 1.012 × 10^−2^, 1.180 × 10^−2^, and 1.349 × 10^−2^ m^2^, respectively. The portable prototype weighs less than four hundred grams.

The photocatalytic membrane reactor was supplied with NH_3_ gas (5%, He balanced) via 1/4 and 1/8 inch stainless steel tubes, with the flow rate controlled by a mass flow meter (MFC). The photocatalytic membrane reactor is described in the Appendix A. The lab-scale photocatalytic membrane reactor was closed in an emetic black box to prevent light from entering. As UV and visible light sources, a UV lamp and a Xenon lamp were used. 

There are two successive steps in the experimental protocol for measuring the photocatalytic degradation characteristics of gaseous ammonia. Figure 1a illustrates the first step. First, the gaseous gas mixture containing NH_3_ was injected continuously through the experimental equipment until the stabilization of the process. The process was considered stabilized when two consecutive measurements of intensity peaks differed by less than one percent, according to gas chromatography (GC, iGC-200A, DS Science, Gwangju, Korea) measurements. The NH_3_ sampling was repeated every five minutes for at least one hour. Figure 1 illustrates the second step (b). The second step is to switch the system from an open situation (see Figure 1a) to an internal circulation situation (see Figure 1b), in which the entire device changes into a batch-type reactor operating in an unsteady state. The lamp installed in the dark box was turned on at the same time as the change in circulation type. The NH_3_ decomposition rate was calculated by dividing the initial concentration (C_0_) by the concentration at time *t* (C*_t_*).

## 3. Results and Discussion

### 3.1. Characterization of Al_2_O_3_ Hollow Fiber Membranes

The Al_2_O_3_ hollow fiber membranes had a 0.3 mm thick wall. The cross-sections obtained by fracturing the Al_2_O_3_ hollow fiber membrane after the high-temperature sintering process (1300 °C; 3 h) are shown in Appendix A. There are two types of pore structure: finger-like pores in the inner edge and sponge-like pores in the outer edge. These findings are consistent with previous research [14,15].

### 3.2. Characterization of Undoped TiO_2_ and N-TiO_2_ Photocatalyst Powders

The XRD patterns of undoped TiO_2_ and N-TiO_2_ photocatalyst powders are shown in Figure 2. The XRD patterns in both cases show diffraction peaks that are well indexed to a tetragonal crystal structure (anatase TiO_2_ structure) as described in JCPDS card No. 21-1272. In addition, the crystallite size of the undoped TiO_2_ and N-TiO_2_ photocatalyst powders was calculated using the Debye–Scherrer equation [17]. 

Taking into account the first most intense peak, the average crystallite size of the undoped TiO_2_ photocatalyst powders obtained from the Scherrer formula is 6.9 nm. The average crystallite size of N-TiO_2_ is 7.1 nm. With the N dopant, the average crystallite size of the TiO_2_ particles increased, in apparent disaccord with expectations [17].

TEM was used to examine the microstructures of the undoped TiO_2_ and N-TiO_2_ photocatalyst powders. The undoped TiO_2_ was structured as monodispersed spherical TiO_2_ particles, as shown in Figure 3a. According to the low magnification TEM image (Figure 3b,c), the average diameter of the individual spherical TiO_2_ particles was around 10 nm, which is consistent with previous studies [12,18]. The high-resolution TEM images in Figure 3e,f show that spherical TiO_2_ particles are mixed with polygonal particles in the N-TiO_2_ photocatalyst powder. The size of the N-TiO_2_ nanoparticles is shown in Figure 3f.

The XPS data of undoped TiO_2_ and N-TiO_2_ powders are shown in Figure 4, which allows us to determine the oxidation states of various elements present in photocatalysts. It is important to note that the N_1s_ peak in the 390–410 eV range is correctly present only in the N-TiO_2_ photocatalyst sample. The N_1s_ spectrum of the N-TiO_2_ photocatalyst, in particular, has a binding energy peak around 399 eV. According to Yalçın et al. [18], the peak represents anionic N substitutionally incorporated in TiO_2_ through O-Ti-N linkages. Furthermore, it can be seen that the O_1s_ peak decreased with the N-doping of the TiO_2_ photocatalyst.

Table 1 summarizes the band gap energy, element analysis (at. percent), and BET surface area (m^2^g^−1^) results. The photocatalyst’s BET surface area was determined using nitrogen adsorption methods. After N-doping, the apparent BET surface area of undoped TiO_2_ increased from 50 ± 2 to 57 ± 2 m^2^g^−1^ [19]. These findings agree completely with those of Bakre et al. [12], where the BET surface area of the N-TiO_2_ sample prepared with urea as a nitrogen source was approximately 42 m^2^g^−1^.

The diffuse reflectance spectrophotometer was used to study the absorbance properties of undoped TiO_2_ and N-TiO_2_ photocatalyst powders, and the results are shown in Figure 5a. Both photocatalysts’ absorption spectra show strong absorption in the UV region, with the absorption edge at about 400 nm. When compared to the undoped TiO_2_ photocatalyst, the absorption band of the N-TiO_2_ photocatalyst is significantly red-shifted. Different authors attribute the observed redshift in the absorption edge to luminescence from localized surface states caused by the recombination of photogenerated electron–hole pairs [12,20,21]. Additionally, the bandgaps are presented in Figure 5b. The bandgap energies for undoped TiO_2_ and N-TiO_2_ photocatalyst were 3.17 eV and 2.36 eV, respectively. The bandgap energies obtained in this work show that, as expected, doping TiO_2_ with N resulted in a decrease in the bandgap.

The samples’ photoluminescence (PL) emission was also measured in order to better understand the behavior of holes and light-generated electrons. Figure 6 shows the PL spectra of undoped TiO_2_ and N-TiO_2_ photocatalyst powders at the wavelengths ranging from 400 to 800 nm. The emission spectra of the two photocatalysts had similar shapes. The emission peaks for undoped TiO_2_ and N-TiO_2_ photocatalyst powders were ~520 nm and ~590 nm, respectively. In detail, for undoped and N-doped TiO_2_, the PL spectra showed a red band at about 590 nm and a green band at around 520 nm, respectively. Defects in N-doped TiO_2_ samples can be assigned to the green PL band [22]. The intensity of the green band decreases with the N doping sample when compared to undoped TIO_2_. Furthermore, the PL intensity of the N-TiO_2_ photocatalyst is lower than that of the un-doped TiO_2_ sample, indicating that these samples have improved charge transfer and effective separation of electron–hole pairs. The PL intensity is known to be directly related to the electron–hole recombination rate, and a low electron–hole recombination rate results in a low PL intensity and then a high photocatalytic performance [23]. Finally, a weaker feature at a wavelength of about 610 nm is usually ascribed to oxygen vacancies and defects on the N doping sample [24,25].

### 3.3. Characterization of Al_2_O_3_-Based Hollow Fiber Membranes Functionalized by Undoped TiO_2_ and N-TiO_2_ Photocatalysts

SEM analysis detected the photocatalyst when the surfaces of the hollow fiber membrane before and after TiO_2_ photocatalytic coating are compared (see Appendix A). Furthermore, because a form such as a low-thickness coating layer does not appear in the SEM cross-sections of the hollow fiber membrane, the amount of photocatalyst coated on the surface of the hollow fiber membrane is presumed small (see Appendix A). The EDS mapping image and EDS line scanning analysis, on the other hand, confirmed the presence of photocatalysts effectively deposited on the Al_2_O_3_-based hollow fiber membrane surfaces. Images of EDS mapping of the surface of Al_2_O_3_-based hollow fiber membranes functionalized by N-TiO_2_ photocatalysts are included in the Appendix A. The EDS line scanning analysis of Al_2_O_3_-based hollow fiber membranes functionalized by N-TiO_2_ photocatalysts is shown in Figure 7. According to the EDS line scanning analysis in Figure 7, the thickness of the N-TiO_2_ photocatalyst coating is in the range of about 1–2 µm.

### 3.4. Photocatalytic Degradation Test of Gaseous Ammonia

The NH_3_ heterogeneous photocatalytic degradation capacity of Al_2_O_3_-based hollow fiber membranes functionalized by undoped TiO_2_ and N-TiO_2_ photocatalysts was studied under different light conditions and photocatalytic membrane reactor configurations at room temperature.

Under ultraviolet light, Figure 8a illustrates the NH_3_ heterogeneous photocatalytic degradation capacity of a photocatalytic membrane reactor consisting of 48 Al_2_O_3_-based hollow fiber membranes functionalized by undoped TiO_2_ and N-TiO_2_ photocatalysts. Under ultraviolet light, both photocatalytic membrane reactors demonstrated good photocatalytic performance, but the initial concentration of gaseous NH_3_ was reduced to zero after only fifteen minutes in the case of Al_2_O_3_-based hollow fiber membranes functionalized by N-TiO_2_ photocatalysts. By contrast, in the case of undoped TiO_2_ photocatalysts, the gaseous NH_3_ completely disappeared after about twenty minutes of ultraviolet light.

The behavior of the Al_2_O_3_-based hollow fiber membranes functionalized by undoped TiO_2_ and N-TiO_2_ photocatalysts under visible light was studied using the same configuration with 48 membranes, and the results are shown in Figure 8b. Under visible light conditions, the photocatalytic membrane reactor made of 48 Al_2_O_3_-based hollow fiber membranes functionalized by N-TiO_2_ photocatalysts performed well. The photocatalytic membrane reactor with Al_2_O_3_-based hollow fiber membranes functionalized by undoped TiO_2_ photocatalyst had a maximum NH_3_ heterogeneous photocatalytic degradation capacity of 30% after 30 min. 

In this work, different configurations were also investigated in order to reduce the number of Al_2_O_3_-based hollow fiber membranes functionalized by N-TiO_2_ photocatalysts. Three new prototype lab-scale photocatalytic membrane reactors with 30, 36, and 42 membranes were developed in our laboratory (see Appendix A). Furthermore, due to the high intensity of the Xenon lamp used in the above setup test case, as well as the increased temperature when used for an extended period of time, a small LED light source (white, blue, and ultraviolet) was installed in the middle of the photocatalytic membrane reactor to be more suitable and compact. By combining three LED lamps in a triangular shape, an internally mounted LED system with a total of nine lamps was developed (see Appendix A). It should be noted that the ideal compact and portable photocatalytic membrane reactor configuration would be to have the light source integrated into the photocatalytic system. This new compact configuration, which places the light source in the middle of the photocatalytic membrane reactor rather than on the outside, was tested for heterogeneous photocatalytic degradation of NH_3_ gas at room temperature (see Appendix A).

The capacity of an NH_3_ heterogeneous photocatalytic membrane reactor consisting of 36 Al_2_O_3_-based hollow fiber membranes functionalized by undoped TiO_2_ and N-TiO_2_ photocatalysts under an LED light source (white, blue, and ultraviolet) was investigated with the goals of (1) discovering relationships between NH_3_ heterogeneous photocatalytic performance and photocatalysts and then (2) determining the optimum operational conditions concerning photocatalysts. The results of this experiment are shown in Appendix A. As could be expected, under LED light sources, the photocatalytic membrane reactor made of 36 Al_2_O_3_-based hollow fiber membranes functionalized by undoped TiO_2_ photocatalyst always showed lower performance than the N-TiO_2_ photocatalyst (see Appendix A). Based on the experimental results obtained with the photocatalytic membrane reactor made with Al_2_O_3_-based hollow fiber membranes functionalized by undoped TiO_2_ and N-TiO_2_ photocatalysts under ultraviolet–visible light (see Figure 8) and LED light sources (see Appendix A), the N-TiO_2_ photocatalysts were used in the following section for the fabrication of different NH_3_ heterogeneous photocatalytic membrane reactors.

The NH_3_ heterogeneous photocatalytic degradation capacity of (a) 30, (b) 36, and (c) 42 Al_2_O_3_-based hollow fiber membranes functionalized by N-TiO_2_ photocatalysts under LED light sources (white, blue, and ultraviolet) is shown in Figure 9. As expected, the number of Al_2_O_3_-based hollow fiber membranes functionalized by N-TiO_2_ photocatalysts increases the NH_3_ heterogeneous photocatalytic degradation capacity. In all cases, the NH_3_ heterogeneous photocatalytic degradation capacity under a white LED light source was only slightly lower than that under a blue or ultraviolet light source.

The prototype lab-scale photocatalytic membrane reactor made up of 42 Al_2_O_3_-based hollow fiber membranes functionalized by N-TiO_2_ photocatalysts can successfully reduce the gaseous NH_3_ in the indoor environment to zero in a short amount of time (fifteen minutes) while maintaining the same performance level. 

To the best of our knowledge, this is the first report on using Al_2_O_3_-based hollow fiber membranes functionalized by N-TiO_2_ photocatalysts to develop a novel compact photocatalytic membrane reactor to completely reduce the NH_3_ pollutant concentration, and its excellent performance is expected to expand our understanding of indoor air pollution.

In conclusion, the heterogeneous photocatalytic degradation of ammonia gas pollutant in a compact and portable photocatalytic membrane reactor with Al_2_O_3_-based hollow fiber membranes functionalized by N-TiO_2_ photocatalysts provides an inexpensive alternative not only to conventional technologies, such as thermal and catalytic oxidation processes, condensation, adsorption on solids, scrubbing and biofiltration [26], but also to alternative methods [1,2].

## 4. Conclusions

Previous research on new processes to eliminate the gaseous NH_3_ pollutant from indoor environments has highlighted the importance of discovering new potentials in the heterogeneous photocatalytic degradation of NH_3_ gas. Our findings demonstrate that Al_2_O_3_-based hollow fiber membranes functionalized with nitrogen-doped titanium dioxide (N-TiO_2_) can be used successfully for heterogeneous photocatalytic NH_3_ gas degradation. 

In summary, Al_2_O_3_-based hollow fiber membranes functionalized by N-TiO_2_ were investigated as an advanced photocatalyst for heterogeneous photocatalytic degradation of the NH_3_ gas pollutant in a novel prototype lab-scale photocatalytic membrane reactor, with a modular design, based on 30, 36, 42, and 48 membranes exposed to UV and visible light irradiation. The NH_3_ gas degradation increases with increasing Al_2_O_3_-based hollow fiber membranes functionalized by N-TiO_2_, according to the heterogeneous photocatalytic test results. Under optimal conditions, the NH_3_ gas degradation efficiency reached 100% after only fifteen minutes.

Overall, the results point to a promising future for NH_3_ heterogeneous photocatalytic degradation in compact and portable photocatalytic membrane reactors with Al_2_O_3_-based hollow fiber membranes functionalized by N-TiO_2_ photocatalysts.

## Figures and Tables

**Figure 1 membranes-12-00693-f001:**
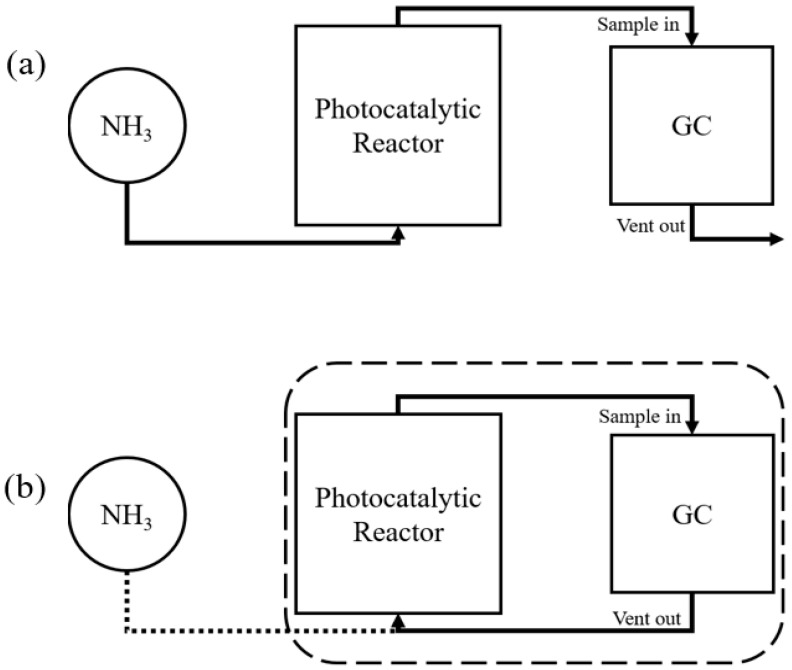
Schematic diagram of the two-step experimental process: (**a**) stabilization and (**b**) batch-type reactor.

**Figure 2 membranes-12-00693-f002:**
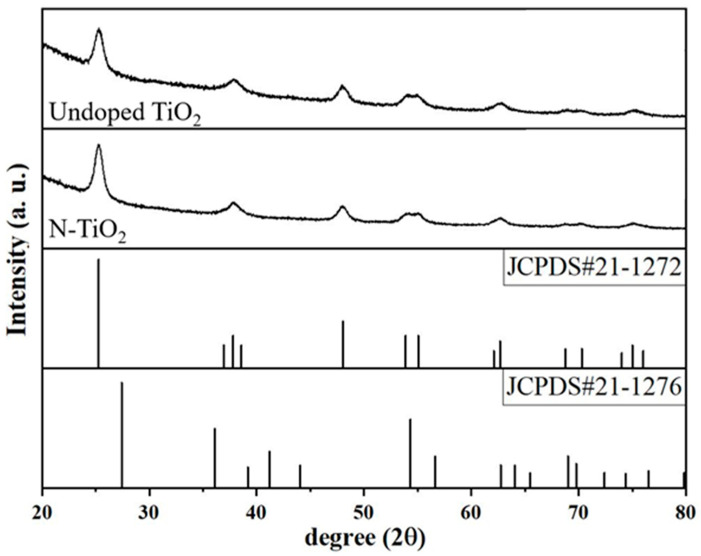
Diffractograms of undoped TiO_2_ and N-TiO_2_ photocatalyst powders. The diffraction profiles of anatase TiO_2_ (JCPDS#21-1272) and rutile TiO_2_ (JCPDS#21-1276) are shown for comparison (bottom patterns).

**Figure 3 membranes-12-00693-f003:**
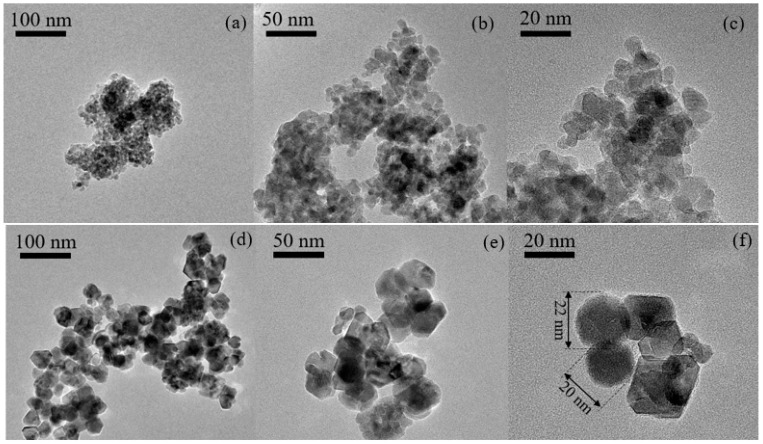
TEM images of (**a**–**c**) undoped TiO_2_ photocatalyst powder and (**d**–**f**) N-TiO_2_ photocatalyst powder.

**Figure 4 membranes-12-00693-f004:**
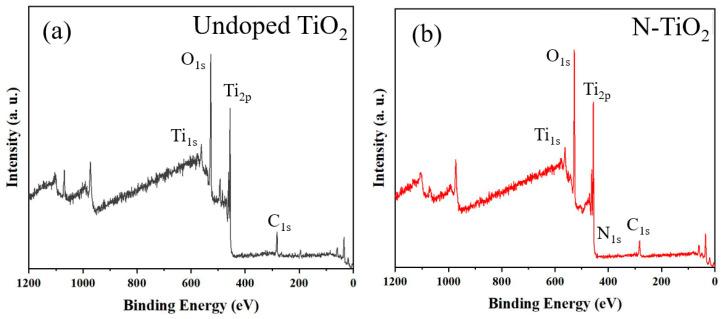
XPS fully scanned spectra of (**a**) undoped TiO_2_ and (**b**) N-TiO_2_ photocatalyst powders. High-resolution XPS spectra for (**c**) Ti_2p_, (**d**) O_1s_, and (**e**) N_1s_.

**Figure 5 membranes-12-00693-f005:**
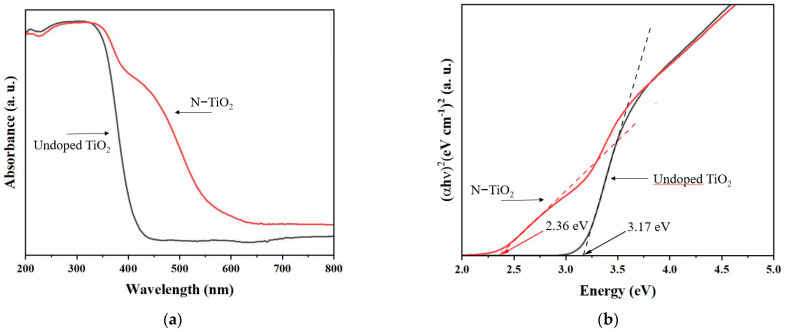
(**a**) UV–vis diffuse reflectance spectra of undoped TiO_2_ and N-TiO_2_ photocatalyst powders. (**b**) The corresponding Kubelka–Munk transformed diffuse reflectance spectra.

**Figure 6 membranes-12-00693-f006:**
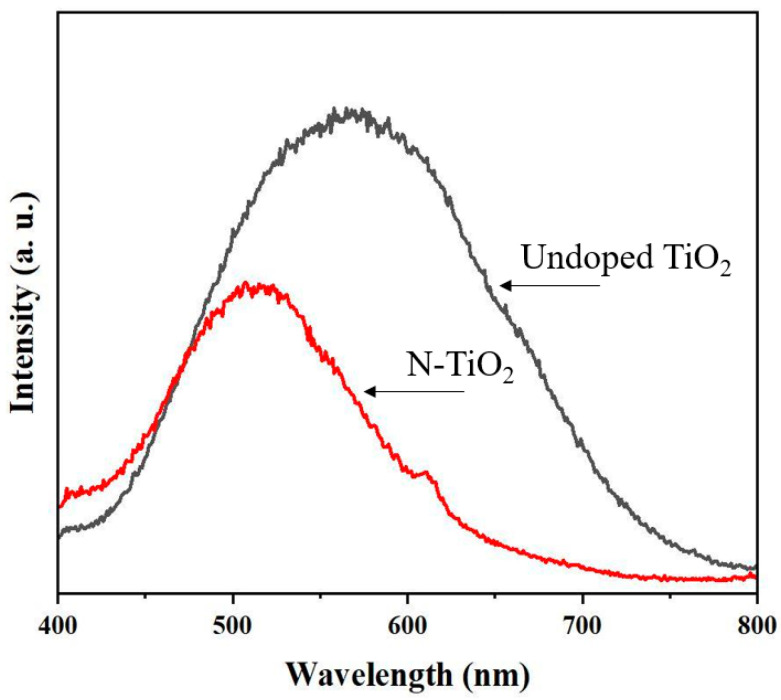
Photoluminescence spectra of undoped TiO_2_ and N-TiO_2_ photocatalyst powders.

**Figure 7 membranes-12-00693-f007:**
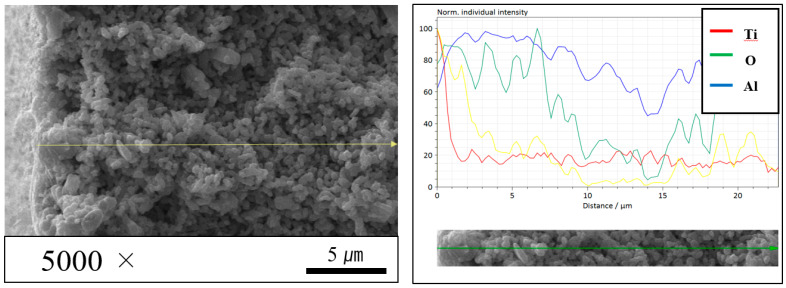
EDS line-scanning analysis across the interface between Al_2_O_3_-based hollow fiber membrane and deposited N-TiO_2_ photocatalyst.

**Figure 8 membranes-12-00693-f008:**
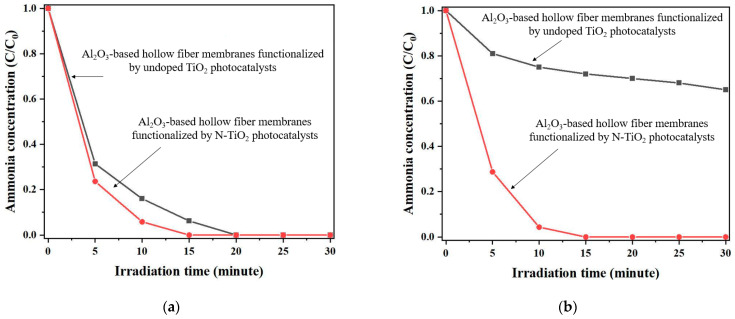
NH_3_ heterogeneous photocatalytic degradation capacity of 48 Al_2_O_3_-based hollow fiber membranes functionalized by undoped TiO_2_ and N-TiO_2_ photocatalysts under (**a**) ultraviolet and (**b**) visible light.

**Figure 9 membranes-12-00693-f009:**
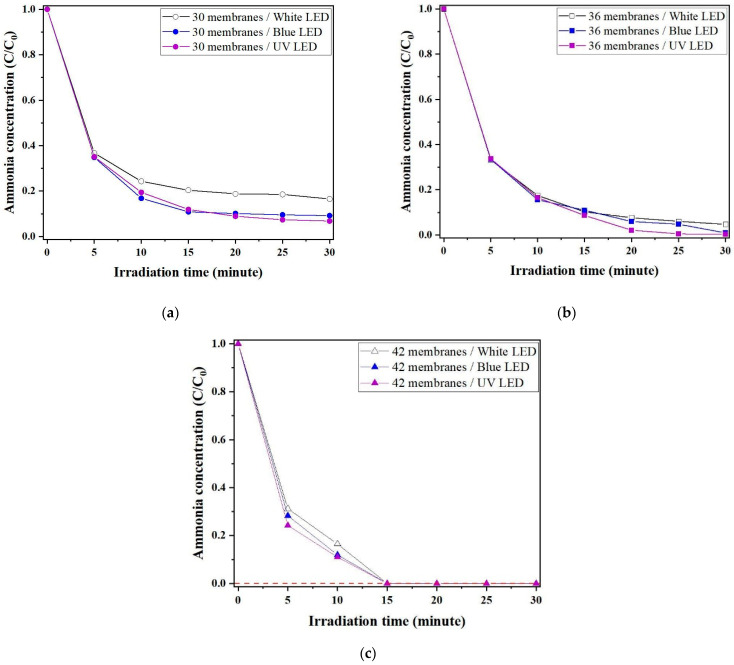
NH_3_ heterogeneous photocatalytic degradation capacity of (**a**) 30, (**b**) 36, and (**c**) 42 Al_2_O_3_-based hollow fiber membranes functionalized by N-TiO_2_ photocatalysts under LED light sources: (**a**) white, (**b**) blue, and (**c**) ultraviolet.

**Table 1 membranes-12-00693-t001:** Summary of the properties of undoped TiO_2_ and (b) N-TiO_2_ photocatalyst powders.

Sample	Band Gap Energy (eV)	Element Amount (at.%)			BET Surface Area (m^2^g^−1^)
		Ti	O	N	
Pure TiO_2_	3.17	26.4	73.6	-	50
N-doped TiO_2_	2.36	27.9	69.2	2.9	57

## Data Availability

Not applicable.

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
