# Peer review of "Al2O3-Based Hollow Fiber Membranes Functionalized by Nitrogen-Doped Titanium Dioxide for Photocatalytic Degradation of Ammonia Gas"

_membranes, 2022, doi:10.3390/membranes12070693_

Round 1
Reviewer 1 Report
In this study, the authors fabricated the Al2O3 based hollow fiber membrane reactor with N-TiO2 for degradation of ammonia gas pollutant. The results are interesting, but the manuscript is just list of experimental results and without much discussion. The below are some issues.
1. in introduction, the authors should explain the advantages of using the membrane reactors to degrade ammonia in air.
2. in experimental section, the effective membrane area should be demonstrated.
3. page 6, the last paragraph, why the BET values increased with N-doping, since the particle size of N-TiO2 is larger than that of undoped TiO2 (as shown in Fig. 3)?
4. Fig. 4, why does the peak of Ti increase and the peak of O decrease with the N doping?
5. Fig. 5a, why the UV-vis diffuse reflectance spectra of N-TiO2 shift the higher wavenumber?
6. The statement of the bandgap energies for undoped TiO2 and N-TiO2 on page 7 is inconsistent with Fig. 5b.
7. Fig. 6, why the PL emission peaks for N-TiO2 sample shift to lower wavenumber?
8. Please explain why the PL intensity of the N-TiO2 samples is lower than that of the undoped TiO2 sample?
9. page 8, section 3.3, one cannot say that no photocatalyst is detected, since the porosity of the membrane surface decreased significantly after N-TiO2 coating, as shown in Fig. S4.
10. page 9, the last sentence, it is not correct to state that “…, but the initial concentration …is reduced to zero … by N-TiO2 photocatalysts”, since both undoped and N-doped catalysts take the same time to reduce the ammonia concentration to zero, as shown in Fig. 8a.
11. Fig. 9, the authors should test the performance of the undoped TiO2 catalyst with white LED and blue LED light sources with 30, 36 and 42 membranes, respectively.
12. What is the main reason for the higher performance of N-doped TiO2 than undoped TiO2?
Reviewer 2 Report
Al2O3-based hollow fiber membranes functionalized by nitrogen-doped titanium dioxide for heterogeneous photocatalytic degradation of ammonia gas pollutant
Status: Minor revision
1. Title is too lengthy, authors should concise it
2. Abstract start with very general info about NH3 which needs to be replace with at least 1-2 lines most relevant to current study
3. Abstract is dispersed, not focused, not clear and no object, authors should re-write it
4. Literature needs improvement in terms of relevancy to the study plus applications. There is a most up to date study, authors should read and included in section-1
• https://doi.org/10.1007/s10973-020-10190-3 [para-1]
• https://doi.org/10.1016/j.jmmm.2016.03.065 [para-1]
• https://www.sciencedirect.com/science/article/pii/S0272884219323739 [para-2]
• https://doi.org/10.1080/19443994.2014.926840 [para-6]
5. Section 2.1, it is recommended to create a table rather to write as para
6. for the composition the authors used previous studies [ref 11 and 12] how it get to be novel then? Need clarifications
7. The difference between pure and n-doped is not much the in fig.4, how N doped hike at 400 BE?
8. What is the difference between fig.8 a & b
9. Conclusion needs to be quantitative and should add the results as well while being concise.
Round 2
Reviewer 1 Report
The authors have addressed all my concerns and I have no more comments.